# Effectiveness of Blood Flow Restriction in Neurological Disorders: A Systematic Review

**DOI:** 10.3390/healthcare10122407

**Published:** 2022-11-30

**Authors:** Maria Jesus Vinolo-Gil, Manuel Rodríguez-Huguet, Francisco Javier Martin-Vega, Cristina Garcia-Munoz, Carolina Lagares-Franco, Ismael Garcia-Campanario

**Affiliations:** 1Department of Nursing and Physiotherapy, University of Cadiz, 11009 Cadiz, Spain; 2Institute for Biomedical Research and Innovation of Cádiz, 11009 Cadiz, Spain; 3Rehabilitation Clinical Management Unit, Interlevels-Intercenters Hospital Puerta del Mar, Hospital Puerto Real, Cadiz Bay-La Janda Health District, 11006 Cadiz, Spain; 4Department of Statistics and Operations Research, University of Cadiz, 11510 Cadiz, Spain; 5PAIDI UCA Group: CTS553, INiBICA Group CO15 Population and Health, Determinants and Interventions, Faculty of Medicine, University of Cadiz, 11003 Cadiz, Spain; 6Group PAIDI UCA CTS391, Faculty of Medicine, University of Cadiz, 11003 Cadiz, Spain

**Keywords:** blood occlusion, blood flow restriction, kaatsu training, neurological disorders

## Abstract

There is scientific evidence that Blood Flow Restriction (BFR) is beneficial in healthy people, the elderly and patients with musculoskeletal disorders. A systematic review was conducted to evaluate the effectiveness of BFR in patients with neurological disorders. The literature search was conducted up until July 2022 in the following databases: PubMed, Web of Science (WOS), Physiotherapy Evidence Database (PEDro), LILACS, Scopus, Cumulative Index of Nursing and Allied Literature Complete (CINAHL), the Cochrane Library and Scientific Electronic Library Online (SciELO). The PEDro scale was used to analyze the methodological quality of the studies, and the Cochrane Collaboration’s tool was employed to evaluate the risk of bias. A total of seven articles were included. BFR seems to be beneficial in neurological disorders. Improvements have been found in sensorimotor function, frequency and step length symmetry, perceived exertion, heart rate and gait speed, walking endurance, fatigue, quality of life, muscles thickness, gluteus density and muscle edema. No improvements were found in lower limb strength or balance. However, results must be taken with caution due to the small number of articles and to the large heterogeneity. More clinical trials are needed. These studies should homogenize the protocols used in larger samples, as well as improve their methodological quality.

## 1. Introduction

In the last few years, research on blood flow restriction (BFR) has gained increasing interest [1,2]. BFR is a system that partly restricts arterial flow and totally restricts venous flow in the exercising musculature during training [3]. This form of exercise with blood occlusion can be traced back to Doctor Yoshiaki Sato in Japan, where it was known as “kaatsu training” [4], consisting of the application of tourniquets or inflatable cuffs to the most proximal portion of the upper and/or lower limbs, which apply an external pressure [5]. BFR has been used in combination with aerobic exercise at a perceived low to moderate intensity, strength exercises where low loads (around 20–40% of 1 RM) are used [6] or merged of electrical stimulation [7].

It has been shown that training with low BFR load (20–30% of 1 repetition maximum, 1 RM) encourages muscular hypertrophy and strength gains comparable to those normally observed after training programs with high load (70–85% of 1 RM) [8].

Low loads may be advantageous, especially for populations that are not capable of lifting near-maximal loads or for whom high loads may be contraindicated, such as in clinical rehabilitation. There is also scientific evidence that the combined application of BFR and electrical stimulation reduces muscle wasting in the legs [6]. A recently published review concluded that the application of the BFR technique can produce short- and medium-term benefits in increasing strength, muscle density and cardiovascular endurance in patients with chronic pathologies [9].

The American College of Sports Medicine recommends resistance training with moderate to high loads to improve strength and muscular endurance [10].

Nevertheless, the use of high muscular tension exercise is often not possible in clinical populations. Hence, BFR could be a rehabilitation tool for clinical populations. In addition, there may be benefits to utilizing BFR training in people with neurological conditions [11].

Neurological diseases are significant disorders that cause varying degrees of disability and loss of productive living [12] with significant health and non-health costs [13]. There are several therapeutic possibilities, including conservative treatment, with the goal of enhancing daily activities, avoiding deformities, preserving joint motion, relieving pain, and reducing muscle tone [14]. While most of the investigations on BFR training have focused on healthy populations [15,16], some case or observational studies have been performed on BFR in patients with neurological problems, such as Parkinson’s disease [17,18], multiple sclerosis [19,20,21,22], spinal cord injury [7] and stroke [23].

To our knowledge, no review exists that analyzes how BFR influences neurological patients. Most of the above studies are limited to postmenopausal women [24], the elderly [25] and musculoskeletal disorders [26]. Therefore, the main objective of this systematic review will be to study the effects caused by BFR in subjects with neurological pathologies and to know if there are any adverse effects when applied in this type of patient.

## 2. Materials and Methods

The Preferred Reporting Items for Systematic Review and Meta-Analysis (PRISMA) model [27] were used for this systematic review. The protocol of this systematic review was registered in the PROSPERO database (CRD42022355426).

### 2.1. Search Strategy

Exhaustive research of studies was conducted up until July 2022 in the following databases: PubMed, Web of Science (WOS), Physiotherapy Evidence Database (PEDro), LILACS, Scopus, Cumulative Index of Nursing and Allied Literature Complete (CINAHL), the Cochrane Library and Scientific Electronic Library Online (SciELO).

“Limb occlusion pressure”, “blood flow restriction”, “blood flow restriction exercise”, “BFR exercise”, “kaatsu training”, “limb occlusion”, “kaatsu”, “vascular restriction”, “cerebral palsy”, “acquired brain injury”, “traumatic brain injury”, “brain injury”, brain disease”, “gait disorders, neurologic”, “hemiplegia”, “diplegia”, “movement disorder”, “stroke”, “physical disability”, “multiple sclerosis”, “neurological disorders”, “spinal cord injury”, “parkinson”, were used as keywords. These were combined with AND and OR.

The research strategy included all available records in any language. Results were filtered to randomized clinical trials (RCTs) or quasi-experimental studies, with no publication date. The detailed research strategies in the different databases are shown in Table 1.

### 2.2. Eligibility Criteria

The studies selected for the systematic review were based on the research question following the format of “PICOS” model [28]: (P) Participants, patients of any age presenting with any neurological disease or pathology; (I) Intervention, blood flow restriction treatment; (C) Comparison, no treatment, placebo or other intervention; Outcome, any clinical variable that could be improved following blood flow restriction treatment; (S) RCT or quasi-experimental studies. Studies analyzing circulatory alterations not related to the BFR intervention were excluded.

### 2.3. Assessment of the Methodological Quality

Regarding the assessment of methodological quality, the scale developed by the Physiotherapy Evidence Database (PEDro) [29] was selected for this study. This scale consists of 11 criteria, with one point awarded for each criterion met, which are (1) selection criteria described; (2) random allocation; (3) allocation concealed; (4) groups at baseline and endpoint were similar; (5) subjects blinded; (6) therapists blinded; (7) blinded evaluators; (8) 85% of participants followed; (9) both intervention and control group results are available even at “intention to treat”; (10) statistical comparisons between groups; (11) point and variability measures. An additional criterion (item 1: selection criteria) related to external validity is added, but this criterion is not used in the calculation of scale scores. Item scores are either (1) or no (0), with a maximum score of 10. Taking into account the specified criteria, studies with a PEDro score of 6 or higher were considered to be evidence Level 1 (6–8: good; 9–10: excellent), and studies with a PEDro score of 5 or lower were considered to be evidence Level 2 (4–5: acceptable; <4: poor) [30].

### 2.4. Selection Process and Data Extraction

The selection process consists of several stages. The first search is performed by combining keywords from different databases. Duplicate articles were then removed using the Rayyan tool (https://www.rayyan.ai/, accessed 15 July 2022). Studies were then selected or excluded. Two reviewers (M.J.V.-G. and I.G.-C.) performed the process of study selection, review, and systematic data extraction. A third reviewer (G.G.-M.) was involved in the process of reaching consensus in the event of a dispute. 

The data extracted from the selected studies were: authors, year of publication, neurological disease or pathology, patient’s characteristics (total number of participants, number of participants in each group), type of intervention, variables, including measurement instruments and results obtained.

### 2.5. Risk of Bias of Included Studies

The risk of bias was calculated for each study selected using the Cochrane Collaboration Tool [31]. It includes six domains: selection bias, performance bias, detection bias, attrition bias, reporting bias, and other bias. Two reviewers (M.J.V.-G. and I.G.-C.) assessed the methodological quality and the risk of bias of the studies. In case of doubt, authors resolved disagreements by consensus and consulting a third author (G.G.-M.) when necessary. This evaluation comprises three terms: “low risk”, “high risk”, and “unclear risk”, and is presented in a table and a chart.

## 3. Results

### 3.1. Selection of Studies

After completing the database search, a total of 275 literatures were identified in combination with various keywords, of which seven studies were finally included in the systematic review [32,33,34,35,36,37,38]. Six of these were randomized clinical trials [32,34,35,36,37,38] and one was a quasi-experimental trial [33]. Figure 1 shows a flowchart of the search process.

A meta-analysis was performed using the EPIDAT program on the four studies [35,36,37,38] that reported quantitative data; nevertheless, due to the clinical, statistical and methodological diversity, it was not possible to perform this meta-analysis.

### 3.2. Data Extraction

#### 3.2.1. Characteristics of the Subjects

There were 129 participants with an average age ranging from 6.6 [36] to 56 years. Ref. [34], of whom 62.2% were female. The largest sample examined is that of Lamberti et al. [34] which had a total of 24 subjects, the smallest being that of Choudhary et al. [32] with 10.

Regarding the neurological pathology of the subjects, 43% of the studies were cerebral palsy [35,36,37] and 28.6% multiple sclerosis [33,34]. There was one article on stroke [32] and one on spinal cord injury [38]. Table 2 shows the main characteristics of the interventions performed in the different studies that make up the present review.

#### 3.2.2. Main Characteristics of the Studies

In all studies, a blood flow restriction intervention was observed in the experimental group. It is noteworthy that in 86% of the trials [32,33,34,35,36,37], BFR is accompanied by exercise training. On the one hand, in four articles, this intervention was accompanied by aerobic exercise [32,33,34,37]. On the other hand, two articles focused on strength exercises [35,36]. Finally, in only one of the articles, no strength or aerobic exercise were practiced, and BFR was supplemented with electrical stimulation placed passively [38].

As for the aerobic training, it was carried out in different ways: with an arm crank ergometry [32], a bicycle [37] or a water-bike [33], or through a low-intensity interval walking exercise [34]. Regarding the article in which no exercise training was performed, a physiotherapy treatment was carried out, specifically neurodevelopmental therapy. Before the training sessions, the type of currents used were biphasic rectangular pulses of 400 μs wide which were released at a frequency of 20 Hz, with a cycle duration of 15 s on and 4 s off [38].

In relation to the treatment time followed in the articles of our review, they ranged from 5 [35,36,37] to 10 weeks [32], with a median treatment duration of 6.4 weeks.

In terms of weekly frequency of sessions, the most commonly used was three sessions per week [33,35,36]. In addition, twice weekly training [34,38], 4 [32] or 5 [37] sessions per week were used.

With respect to the pressure exerted by the cuffs, it was maintained over the period of time during which both strength [35,36] and aerobic training [32,33,34,37] was performed and in the case of Skiba et al. [38], over the time of current administration (15 min). The degree of BFR remained constant throughout the training sessions, although in the article by Lamberti et al. [34], the pressure was released at the end of every six-minute exercise period, and it was maintained at 0 mm Hg for the total duration of the three-minute resting time. The vascular occlusion was achieved using blood pressure cuffs inflated to 60% of brachial artery occlusion [32], to the 30% [34], 40% [35,36,37] or 50% [38] of the systolic blood pressure, with a pressure of 96 ± 10 mm Hg [33], 100 mm Hg and three weeks after 110 mm Hg [35,36,37] or with an average training pressure of 65.2 ± 7 mmHg. The position where the cuffs were placed also varied. They wore around just below the shoulder joint [32], on the most proximal regions of both legs [33,34], between the knee and hip joints [35,36,37] or the thigh immediately below the inguinal crease [38].

In the selected studies, there is a lack of homogeneity with regard to the variables and measurement instruments used. The most studied variable was the muscle thickness using an ultrasonic device [35,36,37,38]. Density [37], white matter muscle area [37] and edema [38] were also studied with the same instrument. Other variables analyzed were gait speed measured by 25-foot walk test [34], walking endurance by 6 min walking test [34], balance through the 14-item Berg Balance scale [34], lower limb strength by the 5-time Sit-to-Stand test [34] or Motricity Index (MI) [32], fatigue evaluated through the fatigue severity scale or the modified fatigue impact scale (MFIS) [34], quality of life with the 36-item short-form health survey (SF-36) [34], sensorimotor function [32] with Fugl-Meyer motor assessment for upper extremity (FM UE), Gross manual dexterity through Box & Block Test (BBT) [32], performance in activities of daily living [32] throughout Barthel Index (BI), step frequency [33] and step length symmetry [33] using a camera.

Concerning the results obtained, in all studies, they were positive. Improvements were found in sensorimotor function [32], frequency and step length symmetry [33], perceived exertion, heart rate and gait speed [34], lower limb strength [34], walking endurance [34], fatigue [34], quality of life [34], muscles thickness [35,36,37,38], gluteus density [37] and muscle edema [38]. No improvements were found in balance [34].

Finally, no adverse effects were identified in any of the trials in this review.

#### 3.2.3. Methodological Quality Assessment

Only one article [34] was of high methodological quality. The rest of them were of acceptable quality. The mean methodological quality of the clinical trials, as measured by the PEDro scale, was five. Table 3 shows the score for each study.

#### 3.2.4. Risk of Bias Assessment

The results for risk of bias are shown in Figure 2. Note that there is a high risk of bias in terms of performance and detection bias as in only one article it was fulfilled [34]. A total of 85.7% of the articles had a low risk of bias in relation with selective reporting [33,34,35,36,37,38]. With respect to other bias all of them were unclear risk (Figure 3). The study conducted by Lamberti et al. [34] presented the lowest risk of bias.

## 4. Discussion

A systematic review has been carried out to synthesize the scientific evidence regarding the use of blood flow restriction in the treatment of neurological disorders.

The most relevant result is that BFR shows improvement in the variables analyzed in all studies, except in balance. According to our findings, there are numerous articles performed on subjects with knee pathology [39,40,41,42], elderly people [33,43,44] or healthy people [45,46] in which an increase in strength has been found with BFR. It should also be borne in mind that in these trials, BFR was accompanied by strength training and in only two of the articles [35,36] in this review was BFR associated with this type of training and in these the strength variable was not measured.

In a study conducted in patients with sporadic inclusion body myositis, a disease is also associated with decreased muscle strength, as is the case with neurological disorders [47], the training protocol with BFR did not improve objective physical function but it had a preventive (retaining effect) effect on the leg muscle strength [48].

Additionally, the meta-analysis carried out by Zhang et al. [49] showed that there was an improvement in the lower limb muscle strength of older adults when low-intensity training was combined with it. Additionally, these authors commented that the duration of training may be an important regulator variable for the positive effect on muscle strength because their results showed that the force gain achieved by BFR when development was longer than 10 weeks is close to that contributed by high intensity training and even significantly superior when training lasts for 16 weeks. The articles in our review had an average duration of 6 weeks, and perhaps this could have influenced the lack of improvement in muscle strength. In any case, strength would be a variable to take into account in neurological problems. Given the light-loading mechanism used in BFR, it can be used in rehabilitation without the elevated levels of joint stress and cardiovascular risk related to hard load training. Furthermore, BFR training shows promise in terms of enhancing function and quality of life [50], variables greatly diminished in neurological patients [51]. Evidence suggests that strength training is useful in the treatment of many of the clinical complaints of this type of subject [52,53]. In the meta-analysis conducted by Cruickshank et al. about strength training, it was found that it was beneficial for fatigue, functional capacity and quality of life in multiple sclerosis [54].This is consistent with the articles in our review that studied BFR in this pathology, although it should be noted that it was accompanied by aerobic exercise rather than strength training [34,55].Other authors who studied the effect of BFR with strength training [19] found that BRF with low-load strength training is well tolerated by people with MS, requiring less muscular effort and causing no pain, compared to high-load training.

Another positive result found in our review was the increase in muscle thickness [35,36,37,38], in some of them BFR was accompanied by aerobic exercise [37], some with strength [35,36] and some with electrotherapy [38]. Regarding the latter therapy, Gorgey et al. also obtained improvements in skeletal muscle size, as well as wrist strength and hand function in people with spinal cord injuries [7]. In terms of the pathologies studied where this improvement was found in our manuscript, it was in cerebral palsy [35,36,37] and spinal cord injury [38].

This outcome improved in 57% of the articles of this systematic review [35,36,37,38]. This is consistent with findings from other trials [51,53,56]. In another investigation by Abe et al. [53], muscle hypertrophy was also induced when BFR was combined with slow walking training. Furthermore, these authors argued that one of the possible mechanisms that could induce it could be insulin-like growth factor type I (IGF-I), growth hormone (GH) and other myogenic regulatory factors.

An additional aspect to take into account, apart from the favorable results, are the adverse effects that BFR may have. None of the articles in our review found any that were in agreement with the majority of the scientific literature [17,18,20]. Nevertheless, one study found adverse effects. The patient had tetraplegia and suffered four incidences of increased blood pressure > 20 mm Hg after BFR training sessions, only one of them being symptomatic. Self-reported mean pain during training ranged from “mild” to “moderate” [57]. However, in the trial conducted by Kjeldsen et al. [58], in patients with incomplete spinal cord injury, the authors found that controlled BFR exercise can be safely performed by individuals with this pathology without added cardiovascular strain or heightened pain.

On the other hand, there are as yet no standard BFR training guidelines. The occlusion pressure, the type of training or associated therapy remains unclear.

Regarding the occlusion pressure, there is no uniformity in the existing literature as in the articles of the present review. In the review of Heitkamp, it was concluded that an occlusion pressure of 150 mmHg may be recommended, and wider cuffs are more efficient than narrower ones [59], although this review was conducted in 2015 on healthy subjects and not in a clinical population. In the study by Stavres et al. on the feasibility of BFR in patients with incomplete spinal cord injury, an occlusion pressure of 125% venous occlusion pressure (28.8% of arterial occlusion pressure (AOP)) was used, commenting that in other research in the general population used a variety of cuff pressures at or above 100% AOP was used [58].

Some research has shown that BFR pressure at 40% arterial occlusion pressure is sufficient for effective acute muscle response [60,61]. However, higher pressure may promote more pain, which may reduce adherence [62].

Lately, it is being used a technique to calculate and prescribe the occlusive stimulus as a percentage of the occlusion pressure of the limb to provide a more individualized prescription but always supervised by experienced professionals. The pressure can be selected based on age, gender, brachial systolic pressure and limb circumference [63]. Therefore, currently it would be advisable to use, in clinical populations, between 40% and 80% of the limb occlusion pressure [39]. The maximum reached in the articles analyzed in this paper was 50% of the systolic blood pressure [38] and 110 mm Hg [35,36,37]. So, these articles are in line with what is found in current scientific evidence.

Concerning the type of training (strength or aerobic) or associated therapy such as FES, it is not possible to determine which one was the most effective, given the large heterogeneity found. BFR has been associated mainly with strength exercises in academic literature, although in our work only two papers addressed it in neurological patients. The rest were in aerobic (3 articles) and one was associated with low-intensity functional electrical stimulation.

Regarding strength training, in order to increase strength, the minimum acceptable intensity for weight training is 65% of a person’s one repetition maximum (1-RM) [6]. For low intensity resistance training, 20–30% of the 1-RM is used [64], which is more beneficial for neurological patients and attenuates stress on the joints and surrounding musculature [65]. This is in concordance with the studies in our review.

On the other hand, aerobic training with BFR has been studied more in healthy people. Regardless of the training mode, it is recommended that exercise training aimed at improving cardiovascular endurance should be performed at intensities greater than 50% of VO2 max, although improvements in VO2 max have also been achieved with low-intensity exercise (<50% of VO2 max) [66]. This variable was not measured in the studies in this review. Other scales were used, more related to functionality, such as sensorimotor function [32], gross manual dexterity [32], performance in activities of daily living [32], step frequency [33], step length symmetry [33], gait speed [34] or walking endurance [34]. It could be interesting to report more objective outcomes in the study of neurologically impaired subjects as well.

Lastly, the last BFR-associated modality found in our review was FES. The combined effect of functional electrical stimulation and BFR has not been thoroughly investigated. The findings of the Skiba et al. trial [38] could be highlighted. Acute changes in muscle density and chronic increases in muscle thickness were identified. FES has been indicated to be effective at stimulating hypertrophy in paralyzed muscles [67]. However, it seems that with the combination, the effects are better. This appears to be in contrast to the fin-dings of research on the quadriceps of healthy people, where neuromuscular electrical stimulation and BFR led to an increase in knee extensor strength but not in muscle mass [68].

From the above, it can be seen that there is little uniformity in the current scientific literature on the benefits of BFR in neurological patients, except for the few adverse effects associated with its use.

Though this paper presents interesting information, some limitations have to be taken into account. Despite an extensive literature review, only seven articles were found with a relatively small number of samples. The quality of the evidence was rated as acceptable and only one article had an evidence level of one [34]. Another limitation was related to the inclusion of studies analyzing patients with different neurological disorders. Furthermore, there is great heterogeneity between studies. It is so broad that a meta-analysis is impossible. There was lack of uniformity in BFR protocols used, time of application, duration, frequency and pressure applied. Moreover, the measuring tools were very heterogeneous and not very objective. It would also be interesting to unify the intervention protocols, as well as to analyze which of the training or electrical stimulation techniques is more indicated to achieve the intended effects. Finally, considering that the participants might have asymmetric limb involvement, have few comorbidities, or be under the effects of pharmacological treatment, it is not known whether this could have any effect on the findings. In addition, another important limitation in drawing conclusions about the efficacy of BFR is that it has not been applied as a unique treatment. In all the articles studied, it was accompanied either by strength or aerobic exercises or by electrotherapy. It would be worthwhile to study its efficacy in studies comparing BFR treatment alone with other therapies.

In general, BFR can be considered an emergent clinical approach to obtain physiological adaptations in individuals who are unable to safely tolerate a high muscular level [11] and with this review, we would like to encourage researchers to study the use of this powerful tool in neurological patients.

## 5. Conclusions

Based on the studies included in this review, it appears that blood flow restriction therapy seems to be beneficial in neurological disorders without adverse effects. Improvements have been found in sensorimotor function, frequency and step length symmetry, perceived exertion, heart rate and gait speed, walking endurance, fatigue, quality of life, muscle thickness, gluteus density, and muscle edema. No improvements were found in balance. However, the results should be treated with caution due to the small number and heterogeneity of articles studied in terms of pathology, type of exercise, or treatment associated with BFR, pressure exerted on the cuffs, intervention time, and duration of therapy. It should also be noted that BFR has not been studied as a single treatment, as it has always been tested in combination with other therapies. It will be necessary to conduct clinical trials that use larger sample sizes and greater homogeneity, as well as to improve their methodological quality. In order to be able to more objectively compare effectiveness compared to other treatments, clinical studies are needed.

## Figures and Tables

**Figure 1 healthcare-10-02407-f001:**
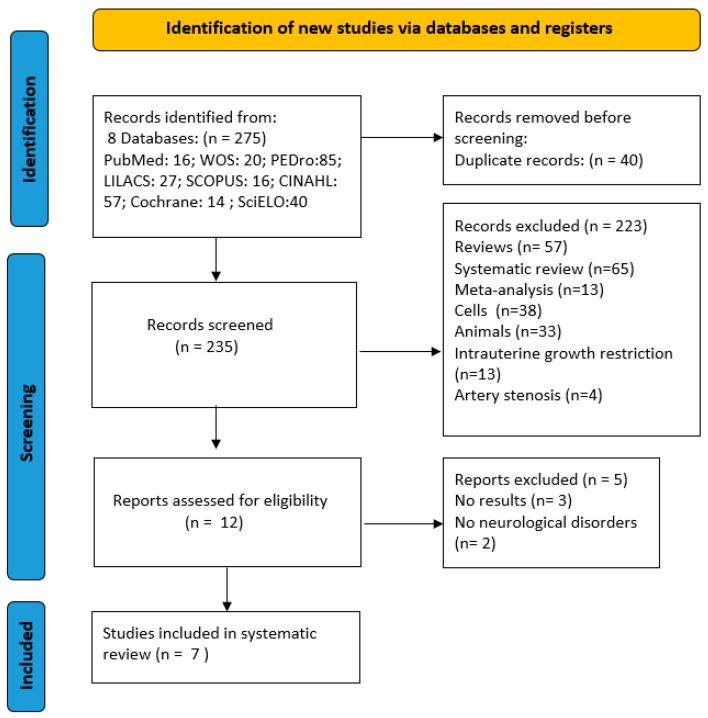
PRISMA 2020 flow diagram.

**Figure 2 healthcare-10-02407-f002:**
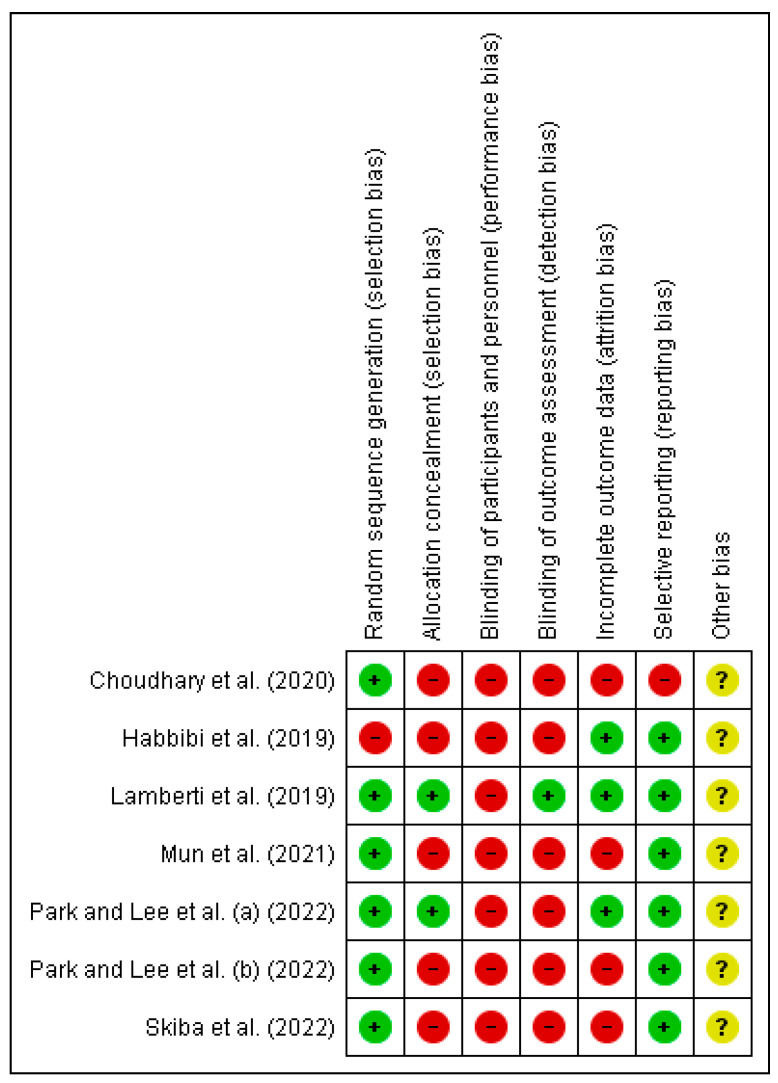
Risk of bias summary.

**Figure 3 healthcare-10-02407-f003:**
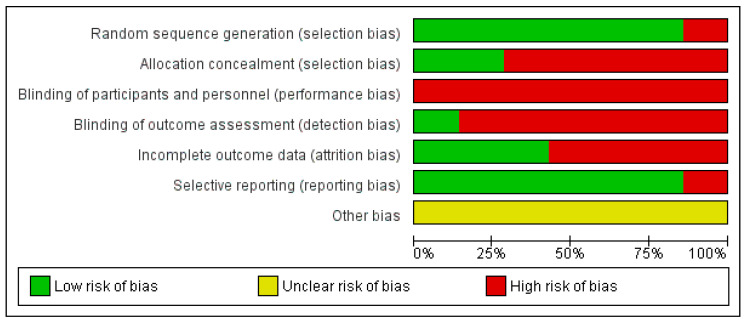
Risk of bias graph.

**Table 1 healthcare-10-02407-t001:** Search combinations.

Databases	Search Strategy
PubMed	(‘Limb occlusion pressure’[Title] OR ‘blood flow restriction’[Title] OR ‘Blood flow restriction exercise’[Title] OR BFR exercise[Title] OR ‘kaatsu training’[Title] OR ‘limb occlusion’[Title] OR ‘blood flow restrict*’[Title] OR ‘vascular restrict*’[Title] OR katsu [Title]))) AND ((‘cerebral palsy’[Title] OR ‘acquired brain injury’[Title] OR ‘traumatic brain injury’[Title] OR ‘brain injury’[Title] OR ‘brain disease’[Title] OR ‘gait disorders, neurologic’[Title] OR ‘hemiplegia’[Title] OR ‘diplegia’[Title] OR ‘movement disorder’[Title] OR ‘stroke’[Title] OR ‘physical disability’[Title] OR ’ multiple sclerosis’[Title] OR ‘neurological disorders’[Title] OR ‘spinal cord injury’[Title] OR ‘Parkinson’[Title])
WOS	TITLE-ABS-KEY (‘Limb occlusion pressure’ OR ‘blood flow restriction’ OR ‘Blood flow restriction exercise’ OR BFR exercise OR ‘kaatsu training’ OR ’ limb occlusion’ OR ‘blood flow restrict*’ OR ‘vascular restrict*’ OR kaatsu) AND (‘cerebral palsy’ OR ‘acquired brain injury’ OR ‘traumatic brain injury’ OR ‘brain injury’ OR ‘brain disease’ OR ‘gait disorders, neurologic’ OR ‘hemiplegia’ OR ‘diplegia’ OR ‘movement disorder’ OR ‘stroke’ OR ‘physical disability’ OR ‘multiple sclerosis’)
PEDro	blood flow restriction
LILACS	blood flow restriction AND neurological disorders in title, abstract, subject
SCOPUS	(KEY (blood AND flow AND restriction) AND TITLE-ABS-KEY (neurological AND disorders))
CINAHL	(Limb occlusion pressure OR blood flow restriction OR Blood flow restriction exercise OR BFR exercise OR kaatsu training OR limb occlusion OR blood flow restrict* OR vascular restrict* OR kaatsu) AND (‘cerebral palsy’ OR ‘acquired brain injury’ OR ‘traumatic brain injury’ OR ‘brain injury’ OR ‘brain disease’ OR ‘gait disorders, neurologic’ OR ‘hemiplegia’ OR ‘diplegia’ OR ‘movement disorder’ OR ‘stroke’ OR ‘physical disability’ OR ‘multiple sclerosis’)
Cochrane Plus	blood flow restriction AND neurological disorders in title abstract keyword
SciELO	blood flow restriction in all indexes

* represents any group of characters, including the absence of characters.

**Table 2 healthcare-10-02407-t002:** Principal studies characteristics.

Author, (Year)/Pathology	Intervention	Outcomes/Measuring Instruments	Results
Choudhary et al. (2020) [32] Stroke	N = 10 CG: n = 5 Aerobic: arm crank ergometry IG: n = 5 arm crank ergometry + BFR warm up: 3 min aerobic training: 12 min (cadence 50–60%)	- sensorimotor function (FM UE) - Gross manual dexterity (BBT) - limb strength (MI) - Performance in activities of daily - living (BI)	- Improvements in FM (F = 17.883, *p* < 0.05). - Positive correlation of FM in IG (r2 = 0.9508) and CG (r2 = 0.9908). - FM scores an IG improvement of 7.29%, compared to CG.
Habibi et al. (2019) [33] Multiple sclerosis	N = 20 G1: n = 10 Aerobic: water bike G2: n = 10 Aerobic: water bike + BFR - Warm up: 5 min - Aerobic training: 18 min of cycling: 3 sets (each for 6 min- 1 min rest at 60–65% of max heart rate) - cooling down: 5 min	- step frequency (camera) - step length symmetry (camera)	Positive changes, no significant differences between groups in the pre- and post-test.
Lamberti et al. (2020) [34] Multiple sclerosis	N = 24 CG: n = 12 Aerobic: low-intensity interval walking exercise 5 bouts of walking and rest period between each. IG: n = 12 Aerobic: Assisted overground walking 40 min + BFR CG/IG: 10 min warm-up and cool-down period/core and stretching exercises.	- Gait speed (T25FW) - walking endurance (6MWD) - balance (BBS) - lower limbs strength (5STS) - fatigue (FSS/MSIS/MFSI) - quality of life (SF-36)	- IG: lower increase in perceived exertion (RPE) (*p* < 0.001) and heart rate (*p* = 0.031) - Gait speed improved significantly in both groups (IG: +13%; CG: +5%) with greater increases in IG at end of the training (*p* = 0.001) and at the follow-up (*p* = 0.041). - Improvements in walking endurance, fatigue and quality of life in IG/IC. No significant differences between IG/CG. - IG: Significant improvements in lower limb strength (d = 0.37), - IG/CG: No significant improvements in balance.
Park and Lee et al. (a) (2022) [35] Cerebral palsy	N = 20 CG: n = 10 conventional strength and balance exercise n = 10 IG: n = 10 conventional strength and balance exercise + BFR upper body strength exercises, abdominal exercises, bridge exercise (5 times/2 min rest in between)	trunk MT: external and internal obliques and gluteus maximus (ultrasonic instrument)	IG: more significant differences in trunk MT (*p* < 0.05).
Mun et al. (2021) [36] Cerebral palsy	N = 20 CG: n = 10 strength exercise IG: n = 10 strength exercise + BFR plantar flexion knee joint extension (4 sets of 3 min training- 2 min rest) 20 min a day	Leg MT: rectus femoris, gastrocnemius, gluteus medius (ultrasonic instrument)	- Rectus femoris, gastrocnemius, and gluteus medius muscles showed significant differences in the groups 5 weeks (*p* < 0.05). - There was a significant difference between the groups in the rectus femoris and gastrocnemius after 5 weeks (*p* < 0.05).
Park and Lee et al. (b) (2022) [37] Cerebral palsy	N = 20 CG: bicycle exercise 30 min N = 10 IG: bicycle exercise 30 min + BFR N = 10	gluteus MT, density, and WAI (ultrasonic instrument)	- CG and IG: significant difference for gluteus MT, density, and WAI (*p* < 0.05). - Between two groups, there was a more significant difference in gluteus MT, density, and WAI than in the control group (*p* < 0.05)
Skiba et al. (2022) [38] Spinal cord injury	N = 16 CG: n = 7 NDT + FES IG: n = 9 NDT + FES + BFR	- MT (ultrasonic instrument) - edema (ultrasonic instrument)	- IG: MT and edema increase compared to CG (*p* < 0.05). - IG: a chronic MT increase after 4 weeks of training (*p* < 0.05). Following 3 weeks of detraining, MT decreased to baseline.

CG: Control group; IG: intervention group; BFR: blood flow restriction therapy; min: minute; FM UE: Fugl-Meyer motor assessment for upper extremity; MI: Motricity Index; BBT: Box & Block Test; BI: Barthel Index; T25FW: timed 25-foot walk test; 6MWD: 6 min walking distance; BBS: Berg Balance Scale; 5STS: 5-time Sit-To-Stand; FSS: Fatigue Severity Scale; MSIS: multiple sclerosis impact scale; MFIS: modified fatigue impact scale; SF-36: 36-item short-form health survey; MT: muscles thickness; WAI: Index of white area; FES: low-intensity functional electrical stimulation; NDT: neurodevelopmental therapy.

**Table 3 healthcare-10-02407-t003:** Methodological quality assessment (PEDro Scale).

Author, (Year)	Item 1	Item 2	Item 3	Item 4	Item 5	Item 6	Item 7	Item 8	Item 9	Item 10	Item 11	Total
Choudhary et al. (2020) [32]	1	1	0	0	0	0	0	1	0	1	1	4/10
Habbibi et al. (2019) [33]	1	0	0	0	0	0	0	1	1	1	0	3/10
Lamberti et al. 2020 [34]	1	1	1	1	0	0	1	1	1	1	1	8/10
Park and Lee et al. (a) (2022) [35]	1	1	0	1	0	0	0	0	1	1	1	5/10
Mun et al. (2021) [36]	1	1	0	1	0	0	0	0	1	1	1	5/10
Park and Lee (b) et al. (2022) [37]	1	1	0	1	0	0	0	0	1	1	1	5/10
Skiba et al. (2022) [38]	1	1	0	1	0	0	0	0	1	1	1	5/10

“0” indicates those items with not scoring; “1“ indicates those items score.

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
