# Peer review of "Effectiveness of Blood Flow Restriction in Neurological Disorders: A Systematic Review"

_healthcare, 2022, doi:10.3390/healthcare10122407_

Round 1
Reviewer 1 Report
This manuscript reviews the use of blood flow restriction training in people with neurologic conditions. There is currently a lack of literature on the use of this modality in people with neurologic conditions therefore this manuscript could potentially provide valuable information regarding this. However, there are major revisions needed to this manuscript. The following is my feedback:
This manuscript is riddled with grammatical errors therefore requires major review and editing of the English language to be suited for publication.
The use of the abbreviation for blood flow restriction (BFR) is used incorrectly numerous times throughout the manuscript where BRF is used.
Lines 81-86 reports the keywords that were used in this systematic review; however these do not match the terms listed in table 1.
Line 149 reports the age ranges and states the ages ranged from 6 to 56 however the upper end is not true. The age of 56 was the mean of that cited article.
Table 2
· There are numerous spelling errors in this table. Additionally, in the intervention column, Lamberti et al (2020) is described inaccurately. For this same study, the results are also reported incorrectly for lower limb strength and balance. The table states there was no significant improvements in lower limb strength or balance however this is not accurate. The IG showed significant differences in 5xSTS test – which is lower limb strength.
· The incorrect first author is listed in Table 2 (Lee et al 2022 – Lee is not the first author).
· Inconsistencies in whether the year of publication is in parentheses.
Lines 168-169: states that no strength or aerobic exercise were practiced and the intervention was just functional electrical stimulation however was this stimulation placed passively? Or did the subjects still complete activities or exercises with the functional electrical stimulation?
Lines 170 – 190: All the dosing parameters would be better if included in Table 2.
Table 3 – inconsistencies in the publication year being in parentheses. Additionally, an incorrect summed score is given for the Lee et al 2022 study (should be 5/10). Also, like the prior comment, this is the incorrect first author.
Line 250 – this statement is incorrect and there were significant improvements in strength.
Line 258 – 261 – this seems out of place as this is talking about a muscle disorder (SIBM) and not a neurologic disorder. What the discussion section is lacking is discussion that delves deeper into use of BFR in neurologic disorders. In my opinion, topics to discuss that would make more sense in this section would be things like why BFR may be beneficial for patients with neurologic disorders, why to consider BFR for people with neurologic disorders, etc.
Lines 326-330: These are not all subjective scales as indicated.
References:
#35 and #37 are the same article.
Author Response
ITEMIZED LIST OF THE REVIEWERS COMMENTS
Manuscript ID: healthcare-1980561
Title: “Effectiveness of blood flow restriction in neurological disorders: a systematic review”.
Dear Reviewer,
We greatly appreciate the editor´s and reviewers’ kind and encouraging comments about our study. We have followed their suggestions, trying to incorporate them into the revised version of our manuscript. Changes suggested by the reviewers have been highlighted in blue. We uploaded the tracked changes manuscript, the clean version revised manuscript and itemized point-by-point response to the reviewer’ comments are presented below.
Editor´s and Reviewers´ comments:
*Reviewer 1
RV: Reviewer
AA: Authors
RV: 1. This manuscript is riddled with grammatical errors therefore requires major review and editing of the English language to be suited for publication.
AA: A review of grammatical errors has been carried out. Thank you for your suggestion.
RV: 2. The use of the abbreviation for blood flow restriction (BFR) is used incorrectly numerous times throughout the manuscript where BRF is used.
AA: Following the reviewer´s recommendation, all errors have been corrected.
RV: 3. Lines 81-86 reports the keywords that were used in this systematic review; however these do not match the terms listed in table 1.
AA: We have changed the keywords in the text to match the search that was actually done. Thank you for your feedback which makes our manuscript better. Page 2, lines 81-87.
RV: 4. Line 149 reports the age ranges and states the ages ranged from 6 to 56 however the upper end is not true. The age of 56 was the mean of that cited article.
AA: We have checked the age range again and corrected it. We had to indicate the average age of the participants as this was not detailed in most of the articles. Thank you very much for your input.
RV: 5. Table 2. There are numerous spelling errors in this table.
AA: Thank you for your comment. Following the reviewer´s recommendation, we have corrected the spelling errors on the table 2.
RV: 6. Table 2 Additionally, in the intervention column, Lamberti et al (2020) is described inaccurately. For this same study, the results are also reported incorrectly for lower limb strength and balance. The table states there was no significant improvements in lower limb strength or balance however this is not accurate. The IG showed significant differences in 5xSTS test – which is lower limb strength.
AA: We have more precisely specified the intervention and the results in this study. There was indeed a significant difference in the 5 sit-to-stand time test (d = 0.37). Thank you very much for noticing the error.
RV: 7. Table 2. The incorrect first author is listed in Table 2 (Lee et al 2022 – Lee is not the first author).
AA: The name was indeed wrong. There has been a confusion with the Korean authors. Thank you again for noticing.
RV: 8. Table 2. Inconsistencies in whether the year of publication is in parentheses.
AA: Our intention was to put all dates in brackets. We have solved it.
RV: 9. Lines 168-169: states that no strength or aerobic exercise were practiced and the intervention was just functional electrical stimulation however was this stimulation placed passively? Or did the subjects still complete activities or exercises with the functional electrical stimulation?
AA: Electrical stimulation was applied passively, without any exercise. This has been added in the text to make it clearer to the reader, thanks to the reviewer's input.
RV: 10. Lines 170 – 190: All the dosing parameters would be better if included in Table 2.
AA: Following the reviewer´s recommendation, we have included the dosing parameters in table 2.
RV: 11. Table 3. Inconsistencies in the publication year being in parentheses.
AA: Our intention was to put all dates in brackets. We have solved it.
RV: 12. Table 3. Additionally, an incorrect summed score is given for the Lee et al 2022 study (should be 5/10). Also, like the prior comment, this is the incorrect first author.
AA: Thank you for your comment. Following the reviewer´s recommendation, we have corrected both errors. We have also had to change the average obtained with PEDro scale, instead of 4.85 is 5: Page 8, line 233.
RV: 13. Line 250 – this statement is incorrect and there were significant improvements in strength.
AA: Thank you for your comment. Following the reviewer´s recommendation, we have corrected this statement.
RV: 14. Line 258 – 261 – this seems out of place as this is talking about a muscle disorder (SIBM) and not a neurologic disorder. What the discussion section is lacking is discussion that delves deeper into use of BFR in neurologic disorders. In my opinion, topics to discuss that would make more sense in this section would be things like why BFR may be beneficial for patients with neurologic disorders, why to consider BFR for people with neurologic disorders, etc.
AA: Without a doubt, their recommendations help us to give more clarity to the discussion part. Thank you very much for your comments. We have added a new paragraph as suggested by the reviewer (Page 12-23, lines 276-282).
RV: 15. Lines 326-330: These are not all subjective scales as indicated.
AA: Thank you for your appreciation. We have removed the word “subjective”.
RV: 16. References: #35 and #37 are the same article…
AA: We are sorry for the mistake. We have corrected it. They are actually the same authors but not the same article. One of the articles studies the trunk muscles and the other article studies the gluteus muscles. But they were wrongly referenced. We are sorry for the mistake. We have corrected it. Besides, the names have been changed in the different tables and in the figure 2.
Please, do not hesitate to contact me, if you require further corrections and information.
Thank you in advance

Reviewer 2 Report
The manuscript by Maria Jesus and coworkers on “Effectiveness of blood flow restriction in neurological disorders: a systematic review”, highlights the potential beneficial effect of BFR in neurological disorders. However, the paper does not provide sufficient data about the positive effect of blood flow restriction (BFR) on patients with neurological disorders. Only a few studies report enhancement in some symptoms such as fatigue, quality of life, heart rate, gait speed, muscle thickness, etc. Additionally, in almost all studies you have reported the BFR is generally accompanied by other activities for instance aerobic exercise and strength exercises. Hence, the positive effect reported can not be attributed only to BFR. It is highly recommended to try to link the BFR tool with each neurodegenerative disorder such as Parkinson's disease and multiple sclerosis as you described in the introduction section and you may also describe the symptoms enhanced after BFR administration.
Author Response
ITEMIZED LIST OF THE REVIEWERS COMMENTS
Manuscript ID: healthcare-1980561
Title: “Effectiveness of blood flow restriction in neurological disorders: a systematic review”.
Dear Reviewer,
We greatly appreciate the editor´s and reviewers’ kind and encouraging comments about our study. We have followed their suggestions, trying to incorporate them into the revised version of our manuscript. Changes suggested by the reviewers have been highlighted in blue. We uploaded the tracked changes manuscript, the clean version revised manuscript and itemized point-by-point response to the reviewer’ comments are presented below.
Editor´s and Reviewers´ comments:
*Reviewer 2
RV: Reviewer
AA: Authors
RV: The manuscript by Maria Jesus and coworkers on “Effectiveness of blood flow restriction in neurological disorders: a systematic review”, highlights the potential beneficial effect of BFR in neurological disorders. However, the paper does not provide sufficient data about the positive effect of blood flow restriction (BFR) on patients with neurological disorders. Only a few studies report enhancement in some symptoms such as fatigue, quality of life, heart rate, gait speed, muscle thickness, etc. Additionally, in almost all studies you have reported the BFR is generally accompanied by other activities for instance aerobic exercise and strength exercises. Hence, the positive effect reported can not be attributed only to BFR. It is highly recommended to try to link the BFR tool with each neurodegenerative disorder such as Parkinson's disease and multiple sclerosis as you described in the introduction section and you may also describe the symptoms enhanced after BFR administration.
AA: First, authors want to thank the modifications suggested by the reviewer and his/her effort to improve our manuscript. We agree with your comment. We have changed our conclusion. A new paragraph about has been added to the limitations section in this regard (Page 14, lines 276-304).
A paragraph has also been added on the most important symptoms that improve (Page 12-13, lines 383-387). Thank you very much for your comments.
Please, do not hesitate to contact me, if you require further corrections and information.
Thank you in advance

Round 2
Reviewer 2 Report
After the revisions made by the authors, the quality of the revised manuscript is highly enhanced. Based on this meta-analysis, blood flow restriction is revealed to have a beneficial role in neurological disorders.